# Pyrazolo-triazolo-pyrimidine Scaffold as a Molecular Passepartout for the Pan-Recognition of Human Adenosine Receptors

**DOI:** 10.3390/biom13111610

**Published:** 2023-11-03

**Authors:** Veronica Salmaso, Margherita Persico, Tatiana Da Ros, Giampiero Spalluto, Sonja Kachler, Karl-Norbert Klotz, Stefano Moro, Stephanie Federico

**Affiliations:** 1Molecular Modeling Section (MMS), Dipartimento di Scienze del Farmaco, Università di Padova, Via Marzolo 5, I-35131 Padova, Italy; stefano.moro@unipd.it; 2Dipartimento di Scienze Chimiche e Farmaceutiche, Università degli Studi di Trieste, Via Licio Giorgieri 1, I-34127 Trieste, Italy; margherita.persico@phd.units.it (M.P.); daros@units.it (T.D.R.); spalluto@units.it (G.S.); 3Institut für Pharmakologie, Universität of Würzburg, Versbacher Str. 9, D-97078 Würzburg, Germany; sonja.kachler@uni-wuerzburg.de (S.K.); klotz@toxi.uni-wuerzburg.de (K.-N.K.)

**Keywords:** adenosine receptor, GPCR, molecular modeling, pyrazolo[4,3-*e*][1,2,4]triazolo[1,5-*c*]pyrimidine, antagonists

## Abstract

Adenosine receptors are largely distributed in our organism and are promising therapeutic targets for the treatment of many pathologies. In this perspective, investigating the structural features of the ligands leading to affinity and/or selectivity is of great interest. In this work, we have focused on a small series of pyrazolo-triazolo-pyrimidine antagonists substituted in positions 2, 5, and N8, where bulky acyl moieties at the N5 position and small alkyl groups at the N8 position are associated with affinity and selectivity at the A_3_ adenosine receptor even if a good affinity toward the A_2B_ adenosine receptor has also been observed. Conversely, a free amino function at the 5 position induces high affinity at the A_2A_ and A_1_ receptors with selectivity vs. the A_3_ subtype. A molecular modeling study suggests that differences in affinity toward A_1_, A_2A_, and A_3_ receptors could be ascribed to two residues: one in the EL2, E168 in human A_2A_/E172 in human A_1_, that is occupied by the hydrophobic residue V169 in the human A_3_ receptor; and the other in TM6, occupied by H250/H251 in human A_2A_ and A_1_ receptors and by a less bulky S247 in the A_3_ receptor. In the end, these findings could help to design new subtype-selective adenosine receptor ligands.

## 1. Introduction

The effects of adenosine are mediated through a family of cell-surface G-protein-coupled receptors, which are currently classified into four adenosine receptor subtypes: A_1_, A_2A_, A_2B_, and A_3_. While the A_1_ and A_3_ receptors interact with G_i_ and G_o_ proteins, the mechanism of the A_2A_ and A_2B_ subtypes is the stimulation of adenylyl cyclase via G_s_ proteins. The consequences of these interactions are a reduction, in the case of A_1_ and A_3_, or an increase, in the case of A_2A_ and A_2B_, in the cAMP levels as the second messenger. In addition, all four subtypes may positively couple to phospholipase C via different G-protein subunits [1,2,3,4]. It has also been demonstrated that adenosine receptors can activate different signal pathways not related to G proteins, like the β-arrestin one, which induces different responses with respect to the G-protein signals [3].

Considering the large distribution of adenosine in the organism, it is quite evident that adenosine receptors could be considered an important target for the treatment of several pathologies. In fact, in the last decades, different classes of potent and selective agonists and antagonists have been reported with the aim of characterizing and better understanding the pathophysiological role of adenosine receptor subtypes and their possible involvement in several disorders [5,6,7,8,9,10,11]. In particular, A_1_ antagonists are investigated for their effect on both cardiovascular and metabolic diseases [7]; A_2A_ and A_2B_ antagonists are attracting increasing attention as cancer immunotherapy agents, while A_2A_ is also under study for the treatment of neurodegenerative disorders (i.e., istradefylline is already approved as adjunctive therapy to levodopa in Parkinson’s disease) [9,10]; and, finally, A_3_ antagonists proved effective in preclinical animal models of brain ischemia and oxygen–glucose deprivation in hippocampal slices [6,7,8,9,10,11]. Despite the undoubtedly therapeutic potential of these ligands, only istradefylline has been approved in the last years. The biggest problem of adenosine receptors is that adenosine signaling is almost ubiquitous in our body, leading to a very broad spectrum of effects that, at a therapeutic level, could lead to several unpleasant side effects [6]. The pyrazolo-triazolo-pyrimidine nucleus substituted in positions 2, 5, N7, and N8 has already been detected as one of the representative scaffolds among human adenosine receptor antagonists [12,13,14].

In fact, it has been widely reported that by modulating the pattern of substitutions, in particular, at positions 5, N7, and N8, various compounds with different profiles of affinity and selectivity toward the four adenosine receptor subtypes have been obtained [15]. The aim of this work is to try to better explain with the help of computational methodologies the influence of these patterns of substitutions on both affinity and selectivity vs. adenosine receptor subtypes. In particular, highlighting the specific residues in the four adenosine receptors that are probably responsible for the observed compounds’ affinity profile could not only help rationalize the obtained results of these compounds and others already reported in the literature, but it could also aid the design of new potent and selective ligands.

## 2. Materials and Methods

### 2.1. Computational Methodologies

The molecular modeling studies have been accomplished on a workstation equipped with a 20-core Intel Core i9-9820X 3.3 GHz processor and by running Ubuntu 20.04 as operating system.

#### 2.1.1. Protein Preparation

X-ray structures with PDB IDs 5UEN [16] and 4EIY [17] were retrieved for hA_1_ AR and hA_2A_ AR. The structures were pre-processed by removing apocytochrome b(562)RIL, replacing IL3, and reverting mutations of the crystal construct into wild type. The structures were then prepared with the Structure Preparation tool of the Molecular Operating Environment (MOE) 2022.02 suite [18] by adding missing atoms/residues/loops and capping N/C-terminals (which were not reconstructed) with acetyl and N-methyl group. Protomeric and tautomeric states were optimized with the Protonate3D tool, and hydrogens were then minimized with Amber 14:EHT force field. Non-protein atoms were removed. The Na^+^ ion, acting as negative allosteric modulator [19], and three coordination water molecules, co-crystallized to hA_2A_ AR on structure 4EIY, were kept for docking studies. Na^+^ and water molecules were also aligned to hA_1_ AR structure, minimized, and kept for further modeling. The influence of Na+ on antagonists’ docking was assessed before, showing how the posing accuracy is affected by the presence of Na^+^ [20].

Structures for hA_2B_ and hA_3_ AR were built by homology modeling using the prepared structures of hA_2A_ and hA_1_ AR, respectively. Extracellular Loop 2 (EL2) (from residue N145 to residue G160) was removed from the A_2A_ AR structure template (4EIY). The models were generated using Prime of the Schrödinger suite [21,22] with knowledge-based setting and keeping ligands, Na^+^, and the three selected water molecules as modeling environment. Non-conserved residues were minimized with OPLS4 force field.

The Ballesteros–Weinstein GPCR numbering scheme, based on counting residues from the most conserved positions in each transmembrane helix (TM), has been employed throughout the manuscript and extracted from the GPCRdb website [23,24].

#### 2.1.2. Molecular Docking

Molecular docking was carried out using Glide [25]. A grid centered on F171/168/173/168 (EL2) and N254/253/254/250 (N6.55) for hA_1_/hA_2A_/hA_2B_/hA_3,_ respectively, was built for all the receptor structures, with inner and outer box dimensions of 10 Å and 30 Å, respectively. Glide-SP [25] was employed for docking, including the “enhance planarity of conjugated pi groups” setting.

In order to enable docking of bulky compounds (**1**–**7**), an induced-fit docking procedure was employed, using compound **4** as reference (the bulkiest compound with reported binding affinity for all AR subtypes). In particular, the induced-fit docking tool of the Schrödinger package was used [26], optimizing side chains within 5 Å of the docking poses and using SP as scoring function for Glide redocking. The trimming option was included for side chains of residues occluding the binding pocket such as E172^EL2^, M177^5.35^, and T270^7.35^ in the case of hA_1_ AR; E169^EL2^, M174^5.35^, H264^EL3^, and M270^7.35^ in the case of hA_2A_ AR; E174^EL2^, M179^5.35^, N266^EL3^, and M272^7.35^ in the case of hA_2B_ AR; and V169^EL2^, M174^5.35^, I253^6.58^, and L264^7.35^ in the case of hA_3_ AR. A refined structure was selected for each receptor with visual inspection (bidentate H-bond with N6.55) and used for Glide-SP docking of compounds **1**–**7**.

Docking results were filtered by excluding poses with positive ligand–receptor van der Waals and electrostatic interaction potential (computed in MOE with Amber 14:EHT force field and assigning PM3 partial charged to ligands). Poses were ranked on the basis of electrostatic interaction potential, and one or two poses per compound were selected, prioritizing diverse conformations showing a bidentate hydrogen bond (or approximate one) with N6.55 using visual inspection.

The selected poses are reported in Appendix A generated using a Python script and the following tools: UCSF Chimera for 3D structure representation [27], MOE for per-residue electrostatic and hydrophobic interaction computation, Gnuplot for heat-map plotting, MEncoder for video assembly.

### 2.2. Chemistry

Synthesis and characterization of compounds **1**–**7** have already been reported in the literature [28,29]; for new experiments on A_2B_ receptor, the same batch of compounds **1**–**4** synthetized before have been used. Briefly, as reported in Figure 1, the 8-alkyl-2-(furan-2-yl)-8*H*-pyrazolo[4,3-*e*][1,2,4]triazolo[1,5-*c*]pyrimidin-5-amine derivatives **8**–**12** were reacted with the corresponding acylchloride in the presence of triethylamine, affording compounds **1**–**7**.

### 2.3. Biology

#### 2.3.1. Binding at Human A_1_, A_2A_, A_2B_, and A_3_ Adenosine Receptors

All pharmacological methods followed the procedures described earlier [30]. In brief, membranes for radioligand binding were prepared from CHO cells stably transfected with human adenosine receptor subtype (obtained as reported in reference [30]) in a two-step procedure. In a first low-speed step (1000× *g*), cell fragments and nuclei were removed. The crude membrane fraction was sedimented from the supernatant at 100,000× *g*. The membrane pellet was resuspended in the buffer used for the respective binding experiments, frozen in liquid nitrogen, and stored at −80 °C.

Compounds were dissolved in DMSO and then diluted to the desired concentration in buffer, and at least 6 different concentrations were tested. DMSO in the final solution never exceeded 2%. For radioligand binding at A_1_ adenosine receptors, 1 nM [^3^H]CCPA was used, whereas 30 and 10 nM [^3^H]NECA were used for A_2A_ and A_3_ receptors, respectively. Non-specific binding of [^3^H]CCPA was determined in the presence of 1 mM theophylline; in the case of [^3^H]NECA, 100 µM R-PIA was used [30,31]. For radioligand binding assay of compounds **1**–**4** at A_2B_ receptors, membranes (20 µg membrane protein) were incubated in a total volume of 200 µL (assay buffer: Tris/HCl pH 7.4, 10 mM MgCl_2_, 0.1% bovine serum albumin) with 10 nM [^3^H]ZM241385. Non-specific binding of [^3^H]ZM241385 was determined in the presence of 0.3 mM NECA. After 3 h at room temperature, samples were filtered and washed as described [31]. K_i_ values from competition experiments were calculated with the program SCTFIT [32] and represented the mean of 3–6 replicates with 95% confidence limits. Binding data for compounds **1**–**12** at the A_1_, A_2A_, and A_3_ adenosine receptors were taken from our previous works [28,29,33]. Binding data toward A_2B_ receptors for compounds **8**–**12** were already reported in a previous work following a different procedure [33].

#### 2.3.2. Adenylyl Cyclase Activity in CHO Cells Expressing hA_2B_ Receptors

Functional studies for the hA_2B_ adenosine receptors were performed using adenylyl cyclase experiments. Minor modifications were carried out on the previously reported procedures [30,34]. In this experimental procedure, the homogenate of hA_2B_-CHO cells was subjected to high-speed centrifugation, and the sedimented membrane pellet was then resuspended in buffer (50 mMTris/HCl pH 7.4) and directly used for the assay. IC_50_ values of antagonists were determined by their concentration-dependent inhibition of NECA-stimulated adenylyl cyclase activity (NECA, 5 μM). About 150,000 cpm of [α-^32^P]ATP was incubated with membranes and the incubation mixture for 20 min without EGTA and NaCl [34]. Hill equation was used to calculate IC_50_ which was the mean of 3–6 replicates. Hill coefficients were near unity.

## 3. Results

As depicted in Table 1, all the compounds showed affinities in the nanomolar range at the four adenosine receptor subtypes with different levels of selectivity. An appropriate discussion regarding the structure–activity relationship profiles of these compounds has been extensively reported [13,14,15,33]; nevertheless, a brief summary may be necessary to better understand the computational studies.

It is quite evident that the simultaneous introduction of bulky acyl moiety at the N5 position and small alkyl groups at the N8 position led to derivatives (e.g., compounds **1**, **2**) with high affinity at the A_3_ adenosine receptor (AR) with good levels of selectivity vs. the other receptor subtypes. The increasing size of the N8 substituent (e.g., compound **5**) led to a significant reduction in affinity at the A_3_ subtype with a reduction in the levels of selectivity. It should be otherwise noted that this pattern of substitution led also to a good affinity at the A_2B_ ARs (compounds **1**–**4**), but the highest affinity was still retained at the A_3_ subtype. Instead, the introduction of thienyl moiety at the N5 position (e.g., compound **6**) induced a reduction in both affinity and selectivity for the A_3_ AR subtype.

In contrast, the derivatives with the free amino function at the 5 position induced a different pattern of affinity. In fact, all the compounds showed high affinity (0.8–2.8 nM) at the A_2A_ AR without selectivity vs. the A_1_ and A_2B_ subtypes while possessing affinity in the high nanomolar range (300–700 nM) at the human A_3_ AR, confirming the fundamental role of the bulky substituent at the N5 position for A_3_ AR recognition and the free amino function for A_2A_/A_1_ interaction.

With the aim of rationalizing this behavior, we performed molecular modeling studies investigating the capability of these compounds to assume a reasonable bound state at the orthosteric binding pocket of the hA_1_, hA_2A_, hA_2B_, and hA_3_ ARs.

The four AR subtypes share a high sequence identity, with values ranging around 40% (Appendix A). In particular, hA_1_ AR shares 46% residues with hA_3_ AR, 43% with hA_2B_ AR, and 38% with hA_2A_ AR, with sequence similarity values above 48% in all cases (see Appendix A). Moreover, hA_2A_ AR has a 45% sequence identity with hA_2B_ AR and 30% with hA_3_ AR, which shares 35% residues with hA_2B_ AR. Most of the dissimilarities are located in the extracellular and intracellular loop regions, with transmembrane (TM) helices having even higher similarities (the TM sequence identity being between a minimum of 45% (A_2B_/A_3_) and a maximum of 66% (A_2A_/A_2B_), see Appendix A).

The availability of X-ray tridimensional (3D) structures for the antagonist-bound A_2A_ and A_1_ ARs in the inactive state provided a starting point for the modeling studies. In particular, structures with PDB code 4EIY [17] and 5UEN [16] were used for the hA_2A_ and hA_1_ ARs, respectively. The choice was driven by prioritizing higher-resolution structures with a low number of mutations in the crystallographic construct. Moreover, the choice of structure 4EIY for A_2A_ AR was made for a receptor co-crystallized with the inverse agonist ZM-241385, characterized by a 7-amino-2-(furan-2-yl)-[1,2,4]triazolo[1,5-*a*][1,3,5]triazine scaffold highly similar to that of the 5-amino-2-(furan-2-yl)-[1,2,4]triazolo[1,5-*c*]pyrimidin of the series of compounds reported here, suggesting the PDB structure, affected by induced-fit phenomena, to be a good candidate for docking studies.

No experimental structure is available for A_3_ AR, and only agonist-bound G-protein-bound cryo-Electron Microscopy structures are available for A_2B_ AR [35,36]. To be consistent throughout the AR subtype comparison and avoid activation-state-dependent differences, the 3D structures of the A_2B_ and A_3_ ARs were modeled using A_2A_ AR’s (see alignment in Appendix A) and A_1_ AR’s (see alignment in Appendix A) inactive-state structures as templates, respectively, which provided the highest similarity to the couples A_1_/A_3_ (46% sequence identity, 65% sequence similarity) and A_2A_/A_2B_ (45% sequence identity, 56% similarity).

Compounds **1**–**12** were docked at the four AR structures using a semi-rigid docking approach (ligand flexible/protein rigid), and one or more reasonable binding modes could be predicted just for compounds **8**–**12** at all receptor subtypes. Two reasonable binding modes were selected for almost all the compounds, prioritizing poses with a bidentate hydrogen bond with N6.55 considering the importance of this residue for antagonist and agonist binding [37,38].

In the case of hA_2A_ AR (Figure 1B, Appendix A), the poses with the best electrostatic interaction potential resembled the X-ray conformation of the 7-amino-2-(furan-2-yl)-[1,2,4]triazolo[1,5-*a*][1,3,5]triazine portion of ZM-241385 (Appendix A). The exocyclic 5-amino group and nitrogen N3 are involved in a bidentate hydrogen bond with N253^6.55^, and the aromatic pyrazolo-triazolo-pyrimidine scaffold participates in a pi–pi stacking with F168 on EL2. The furane ring points deep into the binding pocket, facing H250^6.52^, where a T-shaped pi–pi interaction could occur. The alkyl group at position 8 points toward an area delimited by TM7/1/2/3 and EL2. The 5-amino group is further stabilized by a hydrogen bond with a glutamate on EL2 (E169), which is solvent exposed, but held in proximity to the top of the orthosteric pocket by an ionic interaction with H264^EL3^ in most of the A_2A_ AR X-ray structures. A favorable electrostatic interaction potential is reported for E169 and compounds **8**–**12** (Appendix A). An analogue situation was observed in the case of hA_2B_ AR (Figure 1D and Appendix A).

Also, in the case of hA_1_ AR, hA_2B_ AR, and hA_3_ AR, a ZM-241385-like pose was generated for almost all compounds **8**–**12** (except compound **10** in the case of hA_1_ AR and hA_3_ AR) (Figure 1A,D,C and Appendix A). In the case of hA_1_ AR and hA_2B_ AR (Appendix A, Figure 1A,D), the key residues mentioned above for hA_2A_ AR are all conserved, with F171^EL2^, E172^EL2^, H251^6.52^, and N254^6.55^ of hA_1_ AR and F173^EL2^, E174^EL2^, H251^6.52^, and N254^6.55^ of hA_2B_ AR acting like F168^EL2^, E169^EL2^, H250^6.52^, and N253^6.55^ of hA_2A_ AR. The electrostatic interaction between E172^EL2^ of hA_1_ AR and compounds **8**–**12** is not as evident as that in the case of hA_2A_ AR (Appendix A), and no direct hydrogen bond was observed between the residues and the 5-amino group. However, this is due to the conformation of E172^EL2^ in the X-ray structure, and alternative conformation might be expected in a dynamic solvent-exposed environment. Differently, in the case of hA_3_ AR, the position of E169(hA_2A_)/E172(hA_1_)/E174(hA_2B_) is occupied by a hydrophobic residue, such as V169 (Appendix A). The favorable interaction between negatively charged and polar residues at this position with the free amino group at position 5 is, thus, missing. Moreover, a further difference can be noted: at position 6.52, instead of the Histidine that characterizes all the other subtypes, a less bulky Serine group (S247^6.52^) is present that is incapable of making a T-shaped pi–pi interaction with the furane ring of the compounds. Together with this, a Serine residue (S181^5.42^) also replaces an asparagine at position 5.42, which is conserved in all the other AR subtypes, creating more space at the bottom of the binding pocket.

An alternative reasonable binding mode can be observed for most of compounds **8**–**12** (Appendix A) which is highly ranked in terms of electrostatic interaction potential for the hA_1_ and hA_3_ ARs. The alternative conformation maintains the pi–pi stacking interaction with the phenylalanine on EL2 and the bidentate hydrogen bond with N6.55 on TM6. These poses engage N6.55 through the exocyclic amino group at position 5 and through N6 instead of N3. The 8-alkyl group faces the bottom of the binding pocket, while the furanyl group points toward the extracellular tip of TM2. This pose is also observed in the case of the hA_2A_ and hA_2B_ ARs but seems unfavored compared to the ZM-241385-like one because of a major inward position and consequent steric hindrance of TM2 as compared to the hA_1_ and hA_3_ ARs.

For compounds **1**–**7**, it was not possible to observe the aforementioned binding modes by conducting docking studies. A different approach has been adopted to relieve possible clashes of the receptors with the bulky groups at position N5. The bulkiest compound showing binding affinity at all four receptor subtypes, compound **4**, was selected to optimize the conformation of the binding pocket using induced-fit docking. The refined structures were then used for docking all compounds **1**–**7**. With this procedure, a ZM-241385-like pose was obtained for all the compounds at all four AR subtypes. The exocyclic 5-amino group and nitrogen N3 are engaged in a bidentate hydrogen bond (or approximate one) with N6.55, occupied by N254, N253, N254, and N250 in the hA_1_, hA_2A_, hA_2B_, and hA_3_ ARs, respectively, and the aromatic scaffold is involved in pi–pi interactions with F171, F168, F173, and F168 on EL2 (Figure 2A–D and Appendix A). The N5 substituent increases the contacts with the hydrophobic residues in the binding site in all cases (Appendix A). A hydrogen bond with E172/E169/E174 cannot occur in the case of these compounds with a consequent minor stabilization of the bound state in the hA_1_, hA_2A_, and hA_2B_ ARs. Moreover, the E172^EL2^-H264^EL3^ ionic interaction observed in hA_2A_ AR would also be disrupted, contributing to a loss of ligand stabilization.

## 4. Discussion

In this work, twelve already-known pyrazolo-triazolo-pyrimidines were used to try to explain their profile of affinity and selectivity toward the four adenosine receptor subtypes through the identification of differences and/or similarities in the interaction patterns with the residues of the single subtypes. The four AR subtypes are characterized by high sequence identity in the orthosteric binding pocket; thus, the rationalization of ligand binding selectivity at different AR subtypes is not an easy task. It is reported that extracellular loops, especially EL2, might play a role in the ligand recognition process, especially in determining metastable binding sites along the ligand–receptor binding pathway [39,40,41]. In the current study, the contribution of EL2 or other ELs in the binding process has not been investigated, and, consequently, its role in affecting affinity and selectivity remains to be elucidated. The usage of docking, maintaining full rigidity on the protein structure (regardless of the use of an induced-fit docking technique), does not allow the investigation of the binding kinetics or metastable binding sites along the binding root. Molecular Dynamics (MD) simulations and Supervised Molecular Dynamics (SuMD) [42] could aid in this perspective, but a major limitation still takes place. For A_3_ AR, there is still no availability of 3D experimental structures, and for hA_2B_ AR, the newly released cryo-EM structures in the active state have still not solved the coordinates of EL2 [35,36]. Considering that the reliability of loop modeling is poor, as long as an experimental 3D structure is not released to the scientific community, selectivity studies will still be limited.

Nevertheless, comparing possible binding final states can still be informative, at least to assess the capability of the competitive antagonists to be hosted in the orthosteric sites of ARs.

As mentioned before, the pyrazolo-triazolo-pyrimidines reported in this paper show binding capacity at all AR subtypes. Compounds **8**–**12**, characterized by a free amino group at position 5, show a binding affinity in the nanomolar range in the case of the hA_2A_, hA_2B_, and hA_1_ ARs, while they reach an affinity of hundreds of nanomolar in the case of hA_3_ AR. This trend is inverted for compounds acylated at position N5, with increased affinity and selectivity for the hA_3_ AR subtype. A hypothetical explanation consists of an increased stabilization of the N5 free amino compounds in the hA_1_ and hA_2A_ ARs, thanks to a glutamate residue on EL2 (E172 for hA_1_ and E169 hA_2A_), in proximity of the binding pocket. This stabilizing effect is missing in the case of hA_3_ AR, which bears the hydrophobic V169 at this position. Compounds acylated at position 5 would lose this stabilizing effect in the case of the hA_1_ and hA_2A_ ARs and could hypothetically better fit the more hydrophobic environment created by V169. Unfortunately, this rationalization does not fit for the hA_2B_ AR subtype; even if it bears a glutamate residue on EL2 as with the hA_1_ and hA_2A_ ARs, the binding data for the compounds acylated at position N5 are not so different from those for the compounds with a free amino group at the same position. Thus, this aspect should be further investigated, probably by exploring a hypothetical key role played by residues on EL2 and EL3.

## Data Availability

The data presented in this study are available in this article (and Appendix A).

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
