# Peer review of "Pyrazolo-triazolo-pyrimidine Scaffold as a Molecular Passepartout for the Pan-Recognition of Human Adenosine Receptors"

_biomolecules, 2023, doi:10.3390/biom13111610_

Round 1

Reviewer 1 Report

Comments and Suggestions for Authors
This manuscript comes from a group with long-standing experience in the development of ARs antagonists. Overall, it reads well, and the modelling fits the purposes of rationalizing the selectivity profile displayed by the ligands. My suggestion is to publish in Biomolecules with minor revisions.

In particular:

-       The author should stress the importance of developing ARs (therapeutic applications?) and the difficulties that have hampered clinical trials so far.

METHODS SECTION

-       Line 70: Please provide references to support the role of Na+ as NAM and why to include oi for docking purposes

-       The authors produced homology models of A3 and A2B; A2B is currently available (well resolved in the segments important for this work) and the A3 AlfaFold2 model is also available.  Given that class A GPCR orthosteric sites do not change much between inactive/active states, a quick validation/comparison of the homology models against them sounds reasonable.

-       Explain the GPCR numbering scheme

-       Line 89: the residues trimmed during the Induced fit docking of the cmpd 4 how were modelled back? Were they considered at all?

-       I’m not convinced by the use of Cheng-Prusoff in this specific case. The readout was the cAMP level, which is influenced by the expression and isoform profile of G proteins in the system. Hence, the Ki computed is somehow indicative but probably more insightful alongside a direct assay for GDP release/G alpha dissociation. Can the author indicate a reference for the use of Cheng-Prusoff in the context of GPCRs?

RESULTS:

-       ZM241385 is (correctly in my view) used as a reference, but no figure shows its binding mode

-        Line 172: “the highest potency was still retained at the A3 subtype.” “Affinity“ is probably more suited as no functional assay is reported for A3.

-        Line 204: can the author expand on the “well-known conformational difference between active and inactive states of GPCRs”?

DISCUSSION

“It is reported that extracellular loops, especially EL2, might play
a role in the ligand recognition process, especially in determining meta-stable binding sites along the ligand-receptor binding pathway.” -Please report references to support this claim  
Comments on the Quality of English Language

N/A

Author Response

Reviewer 1

This manuscript comes from a group with long-standing experience in the development of ARs antagonists. Overall, it reads well, and the modelling fits the purposes of rationalizing the selectivity profile displayed by the ligands. My suggestion is to publish in Biomolecules with minor revisions.

In particular:

- The author should stress the importance of developing ARs (therapeutic applications?) and the difficulties that have hampered clinical trials so far.

R: Thank you for the comment. A paragraph reporting the most important possible therapeutic applications of antagonists for the four adenosine receptor subtypes has been added, with a comment on the problems in the development of these compounds.

METHODS SECTION

- Line 70: Please provide references to support the role of Na+ as NAM and why to include oi for docking purposes

R: The reference and the reason for including Na+ in antagonists docking have been added to the manuscript.

- The authors produced homology models of A3 and A2B; A2B is currently available (well resolved in the segments important for this work) and the A3AlfaFold2 model is also available. Given that class AGPCR orthosteric sites do not change much between inactive/active states, a quick validation/comparison of the homology models against them sounds reasonable.

R: The reviewer is right: currently the plethora of experimental and model structures for GPCRs is increasing, and in the case of adenosine receptors, in addition to X-ray and cryo-EM structures of A2A and A1 ARs, recently a cryo-EM structure of the A2B receptor have been released.  As the reviewer highlights, the orthosteric binding site has minor differences between active and inactive state, with major differences affecting an outward movement of the intracellular portion of TM6, a rotation of TM3, and conformational rearrangement of microswitches such as CWxP, DRY, Na+ pocket, NPxxY and PIF motifs. In any case, to be consistent throghout  the AR subtypes comparison and avoid to incur in activation state-dependent differences, we preferred to opt for inactive-like models of A2B and A3 ARs given the availability of A2A and A1 ARs inactive structures.

- Explain the GPCR numbering scheme

R: We thank the reviewer for the hint, since the GPCR numbering scheme was used without mentioning it. Now, reference to the Ballesteros–Weinstein numbering scheme was added to the manuscript.

- Line 89: the residues trimmed during the Induced fit docking of the cmpd 4 how were modelled back? Were they considered at all?

R: We are sorry if the explanation was not clear: the Induced Fit procedure of the Schrodinger package allows the selection of residues which are trimmed and then automatically reversed back during the calculation. In particular, the automatic Schrodinger Induced Fit protocol involves docking at the trimmed receptor, Prime side-chain prediction for the protein-ligand complexes, minimization of the side chains and then redocking. The sentence in the manuscript has been rephrased.

- I’m not convinced by the use of Cheng-Prusoff in this specific case. The readout was the cAMP level, which is influenced by the expression and isoform profile of G proteins in the system. Hence, the Ki computed is somehow indicative but probably more insightful alongside a direct assay for GDP release/Galpha dissociation. Can the author indicate a reference for the use of Cheng-Prusoff in the context of GPCRs?

R: We have published several papers applying Cheng-Prusoff relation to this assay and several companies sell kits for cAMP levels determination and use this equation, but we agree with the reviewer comment, thus we have reported IC50 values for this experiment instead of Kis.

RESULTS:

- ZM241385 is (correctly in my view) used as a reference, but no figure shows its binding mode

R: We have initially omitted a figure showing the experimental binding mode of ZM241385, considering that it is well reported in the literature. For completeness, we have now added a figure in Supporting Information.

- Line 172: “the highest potency was still retained at the A3 subtype.” “Affinity“ is probably more suited as no functional assay is reported for A3.

R: Thank you, we have corrected the sentence.

- Line 204: can the author expand on the “well-known conformational difference between active and inactive states of GPCRs”?

R: The reviewer is right, the reported sentence was not clear and did not explain the reason for not using the active A2B AR structure. The “well-known difference between active and inactive GPCRs” involve an outward movement of TM6, with reduction of the contacts between TM3 and TM6, conformational rearrangement of the toggle switch W6.48 (in CWxP motif in TM6). The movement of W, the “toogle switch”, affects the shape of the orthosteric pocket: for istance, docking of agonists at inactive-structures do not provide crystal-like binding modes. Differently, docking of antagonists at active-structures may provide crystal-like binding modes. In this context, we preferred to be consistent in the use of “inactive”-like states to compare the receptors, and we thus worked with a model of A2B AR (inactive) instead of its active cryo-EM structure. To avoid the sentence to be misleading, we removed it from the manuscript, and we added a brief clarification summarising the reason for the modeling choice.

DISCUSSION

“It is reported that extracellular loops, especially EL2, might play a role in the ligand recognition process, especially in determining meta-stable binding sites along the ligand-receptor binding pathway.” -Please report references to support this claim

R: Few references have been added in this regard.

Reviewer 2 Report

Comments and Suggestions for Authors

In this manuscript Authors aimed to analyse which features of compounds with pyrazolo-triazolo-pyrimidine scaffold possessing the same substituent in position 2 and different moieties in positions 5 and N8 are deciding for the affinity toward subtypes of adenosine receptors. For that purpose considered were 5 compounds with amine group at 5 position and alkyl substituents in position 8 and second group with N-acyl substituents in position 5 and alkyl substituents in position N8. Compounds were obtained, evaluated for activity towards adenosine receptors subtypes (A1, A2A, A2B, A3), for functional profile for hA2B receptor. Discussed were structure activity relationships. Molecular modelling and extended docking studies were performed. In conclusion it was stated that deciding shape for the high activity and selectivity for adenosine A3R have compounds with bulky acyl moieties at N5 position and small alkyl groups at the N8 position. Compounds with unsubstituted amino group at position N5 have shown high affinity at adenosine A2A and A1 receptors with selectivity for adenosine A3R. It is interesting scientific contribution however it needs several supplemented data:

1.      Where are coming from the tested compounds (already known)?, were syntheses repeated? Were compounds obtained, commercial? How was their purity evaluated? Which was the purity of compounds?

2.      In Scheme 1 substituents R1  are once with superscripts e.g. 8  once with subscripts e.g. 9,10…

3.      How were prepared solutions of compounds for in vitro studies?

4.      Were cells with overexpression of subtypes of adenosine receptors obtained by Authors?

5.      Which tests were repeated as described in literature? Which results were only taken from literature?, which reference compounds were used? How many times tests were performed? Where are coming from the used reagents and cells?

It should be clear which results (apart from molecular modelling and docking studies) are new, which were taken from literature or repeated

Author Response

In this manuscript Authors aimed to analyse which features of compounds with pyrazolo-triazolo-pyrimidine scaffold possessing the same substituent in position 2 and different moieties in positions 5 and N8 are deciding for the affinity toward subtypes of adenosine receptors. For that purpose considered were 5 compounds with amine group at 5 position and alkyl substituents in position 8 and second group with N-acyl substituents in position 5 and alkyl substituents in position N8. Compounds were obtained, evaluated for activity towards adenosine receptors subtypes (A1, A2A, A2B, A3), for functional profile for hA2Breceptor. Discussed were structure activity relationships. Molecular modelling and extended docking studies were performed. In conclusion it was stated that deciding shape for the high activity and selectivity for adenosine A3R have compounds with bulky acyl moieties at N5 position and small alkyl groups at the N8 position. Compounds with unsubstituted amino group at position N5 have shown high affinity at adenosine A2A and A1receptors with selectivity for adenosine A3R. It is interesting scientific contribution however it needs several supplemented data:

  1. Where are coming from the tested compounds (already known)?, were syntheses repeated? Were compounds obtained, commercial? How was their purity evaluated? Which was the purity of compounds?

R: This binding data were old unpublished results, thus compounds were evaluated on the same batch evaluated for the binding experiments on the other adenosine receptor subtypes. The following sentence has been added in the chemistry (methods): “for new experiments on A2B receptor the same batch of compounds 1-4 synthetized before have been used”. Concerning the purity of compounds, at that time elemental analysis has been used to assess the purity of synthetized derivatives, and for all compounds values were ±0.4% of the theoretical values for C, H and N. Data are reported in the references cited in the paper:

Michielan, L.; Bolcato, C.; Federico, S.; Cacciari, B.; Bacilieri, M.; Klotz, K.-N.; Kachler, S.; Pastorin, G.; Cardin, R.; Sperduti, A.; et al. Combining selectivity and affinity predictions using an integrated Support Vector Machine (SVM) approach: An alternative tool to discriminate between the human adenosine A2A and A3 receptor pyrazolo-triazolo-pyrimidine antagonists. Bioorganic Med. Chem. 2009, 17.

Michielan, L.; Stephanie, F.; Terfloth, L.; Hristozov, D.; Cacciari, B.; Klotz, K.; Spalluto, G.; Gasteiger, J.; Moro, S. Exploring Potency and Selectivity Receptor Antagonist Profiles Using a Multilabel Classification Approach: The Human Adenosine Receptors as a Key Study. 2009, 2820–2836.

Baraldi, P. G.; Cacciari, B.; Romagnoli, R.; Spalluto, G.; Moro, S.; Klotz, K. N.; Leung, E.; Varani, K.; Gessi, S.; Merighi, S.; et al. Pyrazolo[4,3-e]1,2,4-triazolo[1,5-c]pyrimidine derivatives as highly potent and selective human A(3) adenosine receptor antagonists: influence of the chain at the N(8) pyrazole nitrogen. J. Med. Chem. 2000, 43, 4768–4780.

  1. In Scheme 1 substituents R1 are once with superscripts e.g.8 once with subscripts e.g. 9,10…

R: Thank you, we have converted all subscripts in superscripts

  1. How were prepared solutions of compounds for in vitro studies?

R: Thank you for the comment, a sentence describing sample preparation has been added

  1. Were cells with overexpression of subtypes of adenosine receptors obtained by Authors?

R: Yes, cells were from Prof. Karl-Norbert Klotz, obtained by procedure reported in the cited reference:

Klotz, K.-N.; Hessling, J.; Hegler, J.; Owman, C.; Kull, B.; Fredholm, B.B.; Lohse, M.J. Comparative pharmacology of human adenosine receptor subtypes -- characterization of stably transfected receptors in CHO cells. Naunyn. Schmiedebergs. Arch. Pharmacol. 1998, 357, 1–9.

  1. Which tests were repeated as described in literature? Which results were only taken from literature?, which reference compounds were used? How many times tests were performed? Where are coming from the used reagents and cells?

It should be clear which results (apart from molecular modelling and docking studies) are new, which were taken from literature or repeated

R: specification on which tests are taken from literature and which are new was already reported in the caption of table 1. In order to be clearer, we have added these information also in the methods. In addition, replicates and cells origin have been added. We hope that now the methods resulted clear and we would like to thank the reviewer to have highlighted these inaccuracies.

Reviewer 3 Report

Comments and Suggestions for Authors

1.       The work of a Salmaso and colleagues is an interesting molecular modelling study aimed at investigating possible binding modes rationalizing the pharmacological profile of a variety of adenosine antagonists. The work is particularly of interest given the importance of the adenosine receptors in different physiological and pathological contexts. I thus recommend considering this manuscript for publication in Biomolecules.

2.       The work is interesting and well conducted. The scientific writing is good, and the results are well presented. However, several aspects of the study should be improved. I suggest improving the abstract, by including main study findings and conclusions. Moreover, at the end of the introduction the authors should explain with more details the rational of the work and possible outcomes of study findings. To improve the readability, the aim of the work should also be detailed at the beginning of the discussion. Conclusions alongside some comment on the future perspectives of the study findings should be introduced at the end of the discussion.

3.       The sentences in lines 29-34 are lacking in supporting references. I recommend including these references as a support PMID: 34750517, PMID: 37759787, PMID: 33218074 and PMID: 29848236

4.    Adenosine receptors are also considered as targets for tumor therapy PMID: 34660262 and PMID: 31604539. This important information should be included in lines 37-44

5.    Please separate different sections, eg., lines 58-59, 78-79, 136-137

6.    The source of CHO cells should be included in the methods. Moreover, have the expression levels of all adenosine receptors been previously reported in these cells?

Comments on the Quality of English Language

English is good.

Author Response

  1. The work of a Salmaso and colleagues is an interesting molecular modelling study aimed at investigating possible binding modes rationalizing the pharmacological profile of a variety of adenosine antagonists. The work is particularly of interest given the importance of the adenosine receptors in different physiological and pathological contexts. I thus recommend considering this manuscript for publication in Biomolecules.

R: Thank you

  1. The work is interesting and well conducted. The scientific writing is good, and the results are well presented. However, several aspects of the study should be improved. I suggest improving the abstract, by including main study findings and conclusions. Moreover, at the end of the introduction the authors should explain with more details the rational of the work and possible outcomes of study findings. To improve the readability, the aim of the work should also be detailed at the beginning of the discussion. Conclusions alongside some comment on the future perspectives of the study findings should be introduced at the end of the discussion.

R: We agree with reviewer comments and we hope that additions that we have made in abstract, introduction and discussion have improved the clarity and readability, thus the quality of the work.

  1. The sentences in lines 29-34 are lacking in supporting references. I recommend including these references as a support PMID: 34750517, PMID: 37759787, PMID:33218074 and PMID: 29848236

R: Thank you for this suggestion, we have now included two more references to the three already reported.

  1. Adenosine receptors are also considered as targets for tumor therapy PMID: 34660262 and PMID: 31604539. This important information should be included in lines 37-44

R: Thank you for this suggestion, this paragraph has been extended and both references suggested by reviewer and others have been added.

  1. Please separate different sections, eg., lines 58-59, 78-79,136-137

R: This point has been addressed.

  1. The source of CHO cells should be included in the methods. Moreover, have the expression levels of all adenosine receptors been previously reported in these cells?

R: Thank you for the comment. Please, see response to points 4 and 5 of reviewer 2

Round 2

Reviewer 2 Report

Comments and Suggestions for Authors

Manuscript has been corrected. It can be published in Biomolecules journal